# Understanding System Complexity in the Non-Destructive Testing of Advanced Composite Products †

**Nikita Gandhi** [1,2,*], **Rob Rose** [2], **Anthony J. Croxford** [3] **and Carwyn Ward** [1]

1  Bristol Composites Institute (ACCIS), University of Bristol, Bristol BS8 1TR, UK; c.ward@bristol.ac.uk
2  National Composites Centre, Bristol BS16 7FS, UK; rob.rose@nccuk.com
3  Ultrasonics and Non-Destructive Testing (UNDT), University of Bristol, Bristol BS8 1TR, UK; a.j.croxford@bristol.ac.uk
*  Correspondence: nikita.gandhi@bristol.ac.uk; Tel.: +44-1174568469
†  This paper is an extended version of the conference paper Gandhi, N.; Rose, R.; Croxford, A.; Ward, C. Developing a high-fidelity knowledge base for improvements in the non-destructive testing of advanced composite material products. *Procedia Manuf.* **2020**, *51*, 345–352.

**Abstract:** Non-destructive testing (NDT) is a quality control measure designed to ensure the safety of products according to established variability thresholds. With the development of advanced technologies and a lack of formalised knowledge of the state-of-the-art, the National Composites Centre, Bristol, has identified that the increasing complexity of composite products will lead to some severe inspection challenges. To address the apparent knowledge gap and understand system complexity, a formulaic approach to introduce intelligence and improve the robustness of NDT operations is presented. The systemic development of a high-fidelity knowledge base (KB) involves the establishment of a capability matrix that maps material, component, and defect configuration to the capabilities and limitations of selected detection methods. Population and validation are demonstrated through the experimental testing of reference standards and evaluated against an assessment criteria. System complexity in ultrasonic testing operations focusses on capturing the inherent risks in inspection and the designation of evidence-based path plans for automation platforms. Anticipated deployment of the validated applicability data within the KB will allow for road-mapping of the inspection technique development and will provide opportunities for knowledge-based decision making. Moreover, the KB highlights the need for Design for Inspection, providing measurable data that the methodology should not be ignored.

**Keywords:** non-destructive testing; knowledge management; design for inspection; six sigma

## 1. Introduction

The versatility of composites is attractive for many industrial purposes with the exploitation of inherent material advantages, for example, the ability to combine and tailor properties such as high strength- and modulus-to-weight ratio provides opportunities for improvements in efficiency, sustainability, and operation [1,2]. Whilst the uptake in composites has been affected by the COVID-19 global pandemic, the UK is on track to deliver GBP 10 bn growth in sales from 2016 to 2030 [3].

However, the properties and processing characteristics of composite materials lead to complexity which must be understood for correct usage. Since the material is formulated whilst a part is constructed, the design and manufacturing must be considered as interconnected; therefore, multi-disciplinary design principles and careful processing, material, and parameter selection must be conducted to obtain the required component characteristics. Despite being an involved development process, successful composite designs can provide numerous advantages, such as design flexibility and structural integrity, when compared to conventional metallic structures [4,5].

The complexities realised in design and manufacture often introduce a significant cost, which is amplified by relatively high scrap rates from the generation of non-designed features in a component. These features, or defects, are defined in aerospace applications as irregularities in a material or structure that cause it to depart from its specification as defined during the design process [6]. Defects can occur in the design and manufacturing phases and during the in-service life of the component. Potter et al. [7] discuss the generation of more than 130 defect types and 60 sources of variability for autoclave and resin transfer moulding processes. Whilst some root causes of variability can be found in broadgood material, it is difficult to state the point at which a consequence of variability becomes a defect.

Defects that can be introduced into a structure include delamination/disbonds, fibre waviness, and porosity (void content), amongst others. The occurrence, size, and frequency of the defects depends on the design characteristic and process cycle of the component [8]. Since the properties of composites are strongly influenced by the properties of the constituent materials, their distribution, and interaction amongst them, defects may lead to stress concentrations with the potential to knock down the mechanical performance. They can detrimentally affect the structural integrity of a product, and if unchecked, could lead to catastrophic failure [9]. To protect against this, process verification measures are introduced at various stages in a manufacturing process to identify the unacceptable levels of variability in a part. The performance requirements, or acceptance criteria for defects, are prescribed by designers to dictate the maximum allowable deviation from the designed component and the guidelines of the acceptance criteria should be unambiguous, complete, and testable [8,10]. The level of inspection required for a given feature is driven by its criticality and risk of non-conformance; for example, due to the safety-critical nature of aerospace components, primary structures are subject to a 100% inspection [11].

The field of non-destructive testing (NDT) aims to provide a cost-effective post-cure method to verify components. NDT encompasses a group of specialist methods that involve the identification and characterisation of damage without cutting apart or otherwise altering the original attributes of the tested object, but the use of composites present unique challenges in the application of NDT. Material composition, inhomogeneity, and complex geometries push the limits of what is achievable with respect to the detection of defects. This is particularly true for traditional types of NDT that were developed for metallic structures [12]. For a defect detection method to be suitable, the response for an area of non-conformity must be highly distinguishable from the response for an acceptable region and each acceptable method has its own set of advantages and limitations; therefore, the methods are complementary with a combination needed to ascertain all the relevant information from a defect [13–16]. Moreover, there is no single inspection methodology that fits all methods.

It is evident that a lack of knowledge of the inspection practices for composites exists across a variety of industries and cross-sectoral UK industry events have reinforced this position. Outputs from workshops run by the British Institute of Non-Destructive Testing (BINDT), the National Physical Laboratory, and Composites UK are summarised [17–19]:

- The state-of-the-art for NDT of composites is not completely clear, with the knowledge sharing routes not established.
- Accessibility of understanding what NDT can achieve is an issue within NDT circles, in other areas of the product development process, and with customers. As such, development engineers do not have a justification for NDT investment.
- NDT has not been able to adapt to industry requirements, which leads to a problematic deployment of methods. Ineffective use leads to delays at great expense.
- Qualification and standardisation of NDT methods has not been formalised across industries.

In the aerospace industry, defect criteria and failure constraints captured decades ago are still being used in the modern-day acceptance criteria [20]. This is of concern when the employed materials have evolved and do not necessarily provide the same response to

verification techniques as previously established. In the automotive and marine industries, the requirements for inspection are driven by different factors, such as a short-life and stiffness, but a lack of experience and guidance have limited the use of NDT methods. As a result, NDT operations are struggling to overcome the bottleneck in composites manufacturing that result from the current delays in inspection.

Proposed routes from these workshops to tackle the knowledge deficit and knowledge gap include: linking inspections to structural integrity through structural analysis defect criticality assessment to determine the impact of detected defects, technology mapping to establish a state-of-the-art map of appropriate inspection technologies with respect to the characteristics of composite components, and the integration of NDT 4.0 to allow for interoperability between NDT and manufacturing processes for the optimisation and prediction of inspection practices [17–19]. A central theme that emerged from the discussions involved utilising NDT as a learning tool to increase confidence; however, each of the routes proposed requires consistent funding and a central partnership dedicated to best practice, with stakeholders in key NDT and engineering organisations.

The National Composites Centre (NCC), Bristol, UK, a research and development centre for composites design and manufacture, has commenced a research programme which aims to tackle some of the challenges in NDT. To support the existing and future manufacturing demands, a requirement for large-scale automated NDT capabilities has been identified. The Composites Integrity Verification Cell (CIVC), shown in Figure 1, aims to provide a fast, reliable, high quality automated inspection of large, complex-shaped components; the multi-method, single platform system seeks to optimise complex inspection. In deployment, the system would allow for faster implementation and qualification of automated inspection methodologies and to improve the coverage, sensitivity, and repeatability of NDT.

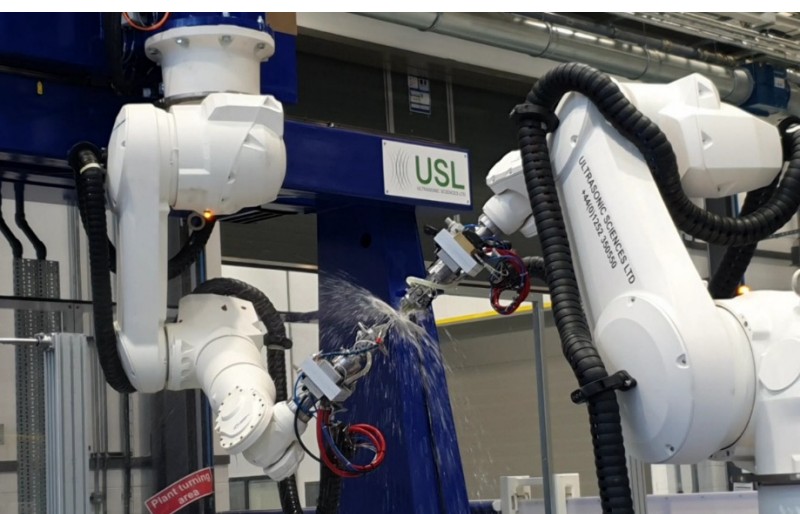

**Figure 1.** NCC's CIVC in operation (representative image).

Whilst there is great potential for CIVC to assist in responding to changing industry requirements, learning risks are a prohibiting factor for effective deployment. A slow pace of innovation and development has been reported for CIVC, delaying the deployment of prescribed system capabilities. Additionally, there is a fundamental lack of understanding of the system capabilities and critical factors that must be integrated for system intelligence.

Drawing influence from cross-sector NDT workshops and CIVC needs, it is identified that the gap in understanding the current state of industrial NDT is a developmental priority. Additionally, the lack of appreciation of and the misconceptions about the benefit of process verification and quality assurance have led to the improper use of technologies; therefore, this work seeks to address the NDT knowledge-deficit to future-proof inspection

activities, and subsequently improve manufacturing as a closed loop through Design for X principles. This is achieved through a Lean Six Sigma (LSS) framework with two aims:

- The development of knowledge management (KM) activities for the documentation of inspection results and technical know-how in the form of a high-fidelity knowledge base (KB).
- Establish a framework for exploiting NDT knowledge to inform the design for inspection of components.

Section 2 describes the use of selected NDT methods and techniques and evaluates the current state of KM within engineering and inspection operations. Section 3 presents the quality improvement framework employed in this study. The seven developmental activities include the formulation, testing, and population of the KB, establishment of inspection risks, and demonstration of system complexity. Activity discussion and suggestions for the future of NDT for composites are explored in Sections 4 and 5.

## 2. Current State of NDT for Composites Operations

As an inherent part of the manufacturing chain, NDT is a complex, multi-parameter process, that is specific to each application and must be carried out by a skilled operator trained to an industrial standard. NDT operations are strongly dependent on both controllable and non-controlled parameters and variances, as demonstrated in the process flow in Figure 2.

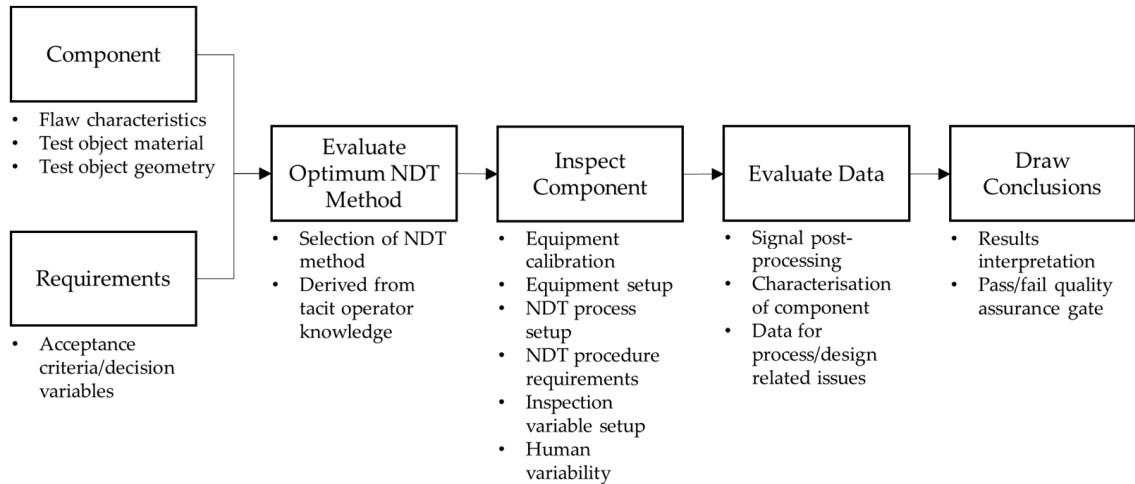

**Figure 2.** Multi-parameter NDT system flowchart.

The success of an NDT procedure is measured on the ability to provide accurate, reliable, and precise results for quality control measures to assure the safe operation of a component. Components that undergo NDT must typically conform to high safety assurance, with a failure to detect potentially leading to significant damage to the component and the consumer. Standards, certification, and qualification procedures exist for this purpose; to attempt to remove an element of uncertainty from a high-risk operation [21]. Within the standards, the acceptable limits of variability are defined for NDT data evaluation purposes, but while these limits should be based on knowledge of the component design and manufacturing route, an understanding of the resolution and reliability of the inspection methods is also critical to the definition. The reliability result can be expressed in terms of reproducibility, repeatability, and capability [22]. There are four possible outcomes from the inspection of a component [23]:

- True positive: item is flawed and is detected by the NDT method.
- False positive: no flaw exists, but the NDT method indicates a flaw is present.
- True negative: no flaw exists, and the NDT method does not indicate a flaw is present.
- False negative: item is flawed, but the NDT method does not detect it.

There are two formal metrics for predicting the NDT capability. The probability of detection (POD) aims to quantify the probability of detecting a specific flaw through experimental and statistical model-assisted methods to plot a POD curve [24]; however, this method is based on metallic structures and does not take three-dimensional damage, as is sometimes found in composite structures, into account [25]. The receiver operator characteristic (ROC) aims to evaluate accuracy and sensitivity through a comparison of the POD and false positive rate for the defects of critical dimension. ROC accounts for signal-to-noise ratios that affect the detectability of flaws and the probability of a false alarm, defining a sensitivity threshold [26].

Reliability has been suggested to be dependent on three components: human factors, application parameters, and intrinsic capability. It is understood from studies by Bertovic et al. [27–29] and Krishnamoorthy [30] that the reliability of a test is dependent on human factors; experience, knowledge, and the skills of the operator contribute greatly [31]. Human error is included within this component, where unreliable human performance is attributed to inadequate interactions between a human and a situation [32]. External influences of accessibility and environmental conditions are encompassed by application parameters, whereas intrinsic capability refers to the physical-technical process that underpins an inspection technology, and defect and geometry configuration. This directly assesses the capabilities and limitations of the inspection system. Since NDT is expected to provide accurate results, the number of false readings should be kept to a minimum; however, this may not be practically achievable. Any false results contribute to increased costs, either through unneeded concessions or safety-related failures. Reliability analyses, solely based on intrinsic capability, may not represent full system capabilities as actual performance can be diminished by an interaction of factors that have not been considered in the POD or ROC metrics [27].

### 2.1. Ultrasonic Testing

Ultrasonic testing (UT) is a commonly used method in the aerospace industry and is based on the transmission of high frequency sound waves through a test object using a coupling medium. Material properties can be determined through monitoring the loss of original amplitude, or energy, in the response pulse [33]. This pulse is dependent on how the ultrasonic wave propagates through the composite and is a function of the beam incidence angle, wave velocity and material density, and how it interacts with the interfaces, discontinuities, or defects [33]. The loss of signal is referred to as attenuation and can be due to reflection, scattering at the internal interfaces, and absorption in the bulk material. This loss is significant in composites, especially at high frequencies, due to the inherent material inhomogeneity. UT can be deployed with a single element unit as well as through a phased array system that uses multiple elements of an array to transmit and receive signals. In recent years, there has been a rapid increase in the use of arrays over conventional transducers, attributed to greater flexibility, imaging performance, and signal stability [34]. Applications of UT have been reported in evaluating the effects of fatigue and damage tolerance on aircraft fuselage structures, detecting cracks, and detecting disbonds in wind turbine blades [35–37]. The two conventional approaches to UT are chosen dependent on the specific application, with careful consideration given to the material and geometry specification and quality control requirements [38–40]:

- Pulse-echo UT: Based on the detection of echoes produced when a transducer/receiver unit introduces a pulse, which is then reflected by a discontinuity in the component. Echoes provide information on the flaw location, depth and size.
- Through-transmission UT: Based on the depletion of amplitude at separate transducers either side of the part. It only provides information on the flaw location and size.

### 2.2. Thermographic Testing

Thermographic testing (IRT) is an emerging non-contact method gaining traction for the fast inspection of composites. Thermal waves within a structure are introduced with

an external stimulus and the thermal gradients are monitored using an infrared camera. Heat diffusion over an irregularity will be obstructed and will differ as compared to the rest of the structure. Defects are typically detected as blurry indications such that the sizing is difficult to accurately determine [41,42]. Depth measurements require knowledge of the thermal characteristics of the host material. As an empirical rule, the radius of the smallest detectable defect should be at least one to two times larger than its depth under the surface. As such, IRT is best for surface or shallow sub-surface defects. IRT has been used to determine delamination defects in wind turbine blades, barely visible impact damage, and bond lines in car rims [43–45]. The IRT method encompasses three techniques [46–48]:

- Pulse IRT: a component is heated with an optical flash exposing the material to a heat impulse, the duration of which can range from milliseconds to seconds.
- Lock-in IRT: Based on the application of modulated optical radiation which propagates as a modulated thermal wave. The amplitude and phase shifts are observed in the response wave.
- Transient IRT: heating of the target surface by a constant low intensity heat flux for a duration of time ranging from seconds to minutes.

### 2.3. Knowledge Management in NDT

Engineering product development has changed over the past decade as a result of increasing product complexity. To enable the knowledge-intensive integrated product development activity, the design and manufacture of new composite components require cross-functional linkages and collaboration [49,50]. This demonstrates a need for KM. At its core, KM involves the management of knowledge artefacts, human sources of knowledge, and processes of how knowledge is generated and applied, and it requires a robust knowledge management system (KMS) to implement it [51]. Since design choices significantly affect other manufacturing parameters with costs escalating further through the design process, attempts have been made to integrate KMSs in the form of concurrent engineering, or Design for Manufacture (DFM) methodologies [52,53].

There is an observed interest in DFM for composites, with methodologies focusing on both the frameworks and sub-processes or tools. An early publication by Gandhi et al. [54] proposed a simultaneous incorporation of key variables in component design and manufacture, whilst a more recent study by Crowley [55] evaluates the transformation of the typical waterfall process into a cyclic learning process. Tools for the improved understanding of the manufacturing process and constituent selection are developed through the DFM platforms LayupRITE and PROSEL [56–59]. Case studies of the DFM strategies in industry are limited; however, a strategy deployed by SAAB Aerostructures and summarised by Andersson et al. [60] aims to establish a process flow to structure the product development process [61].

According to academic engagement, KM in NDT is not a heavily researched field; lack of uptake is evident in the slow pace of innovation with decentralised learning across industries and institutions and the incremental nature of the inspection method maturity. The spread of knowledge is often delayed through a limited means of translation or accessibility from state-of-the-art academic papers to industrial applications. There is some evidence of industrial associations for the source and dissemination of NDT innovations, where large corporation members provide the funding for collaborative projects; however, the visibility of developments is restricted to members only [62].

Despite this, a new interest in inspection strategy has begun to gain traction in the field of Design for Inspection (DFI). This presents a methodology that would provide recommendations on how to design a product so that effective inspection can be achieved [63–65]. A rudimentary decision-tool is discussed for complex NDT operations to support the designation of an interrogation method, whilst an inspectability index is suggested for assessment component suitability [66–68]; however, none of the instances that have reported the development or use of DFI are composites-related or have a clear direction for improvement. For the optimisation of inspections, a formalised model of testing the capabilities,

phenomena of interest, and requirements with respect to the composite components that is accessible across multiple industrial sectors is necessary. This presents opportunities for the development of KM in improving the robustness of NDT for composites.

## 3. Defining NDT System Complexity

This study has adopted a hybrid LSS Define–Measure–Analyse–Improve–Verify (DMAIV) framework, modified by the authors from conventional LSS structures, for the quality improvement opportunity for NDT operations, combining a proactive and reactive LSS methodology for the addition of a new process to a current process. While typically linear, the adopted process is iterative and concurrent, and is better demonstrated with an agile approach, shown in Figure 3. The reorganisation in structure has been performed to allow for the novel nature of the developmental activities with multiple iterations required to determine and prove out the KB structure.

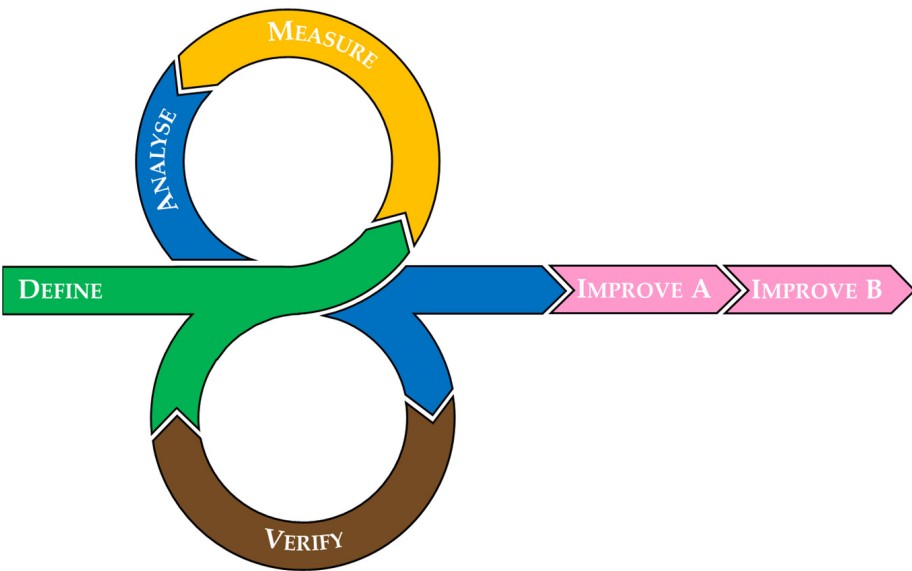

**Figure 3.** Adopted DMAIV project framework with an agile approach.

The selection process for the LSS tools is highly dependent on the application, with the combination shown in Table 1. Utilised tools that are well-established in LSS activities fall into three groups: customer-based, visualisation-based, and statistical-based, and were selected through a review of industrial case studies, prior experience, and trial-and-error.

**Table 1.** Designation of LSS tools to DMAIV framework.

| Define | Measure | Analyse | Improve | Verify |
|---|---|---|---|---|
| Project charter | Design of Experiments | Process capability measurement | Root-cause analysis | Multi-variation assessment |
| Value-stream mapping | Process capability measurement | Prioritisation matrix | Poka-Yoke | |

### 3.1. Define

Industrial and academic influences have established a need for technology mapping in the form of a KB. By mapping as-manufactured composite material, component, and defect configuration to selected NDT methods in a database, a comprehensive understanding of the state-of-the-art capabilities and limitations can be realised.

The Suppliers, Inputs, Process, Outputs, Customers (SIPOC) tool in Figure 4 has been used to evaluate the inputs and deliverables required for effective NDT. Focusing primarily on the inputs, process, and outputs to the system, key system enablers for

inspection can be identified and are selected as the foundation of the capability matrix. The enablers are organised into two groups: the component configuration and NDT process parameters. The component configuration includes design choices for geometry, material characteristics, and the manufacturing processing route. The NDT process parameters include the key performance variables (KPVs) that an operator is required to input in either the physical or equipment setup to maximise the valuable output. Optimal characterisation of the component enables the interrogation and acquisition of as much component quality information as possible.

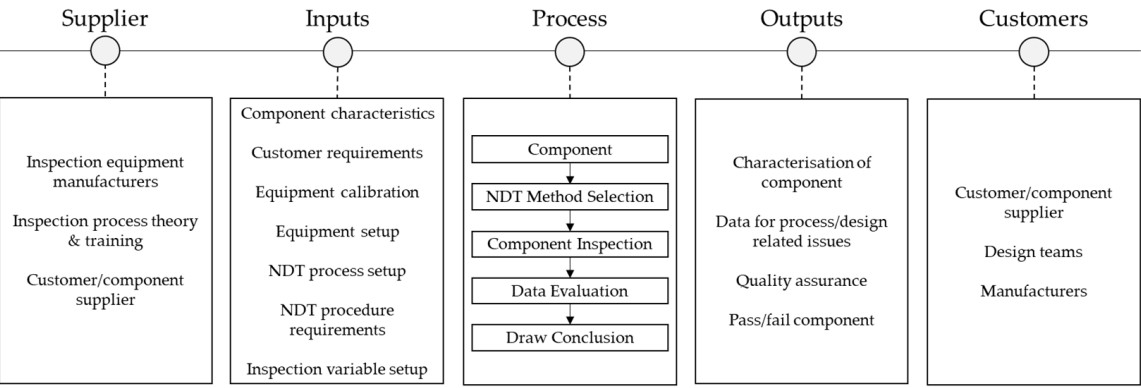

**Figure 4.** SIPOC diagram mapping an inspection process.

### 3.2. Measure

The structure of the capability matrix has been established with the determination of the capabilities of selected NDT methods with respect to component configurations in the previous phase. The two NDT methods selected, UT and IRT, are representative of an established and emerging method to prove out the matrix. The KB must then be populated with current state data through the design and manufacture of reference standards that incorporate component configurations through a Design for Experiments approach.

Down-selected carbon fibre (CFRP) configurations, representative of typical geometries within the composites industry, are categorised by geometry, source materials (preimpregnated, prepreg, or dry fibre material), specimen architecture (monolithic or sandwich), and defect type (delamination or inclusion-type defects). Within specific NDT rules, the material choices, geometrical configuration, and artificial defect placement are designed with the aim of reducing extraneous sources of variability. To cover the 24 configurations associated with monolithic structures and 10 sandwich configurations, four monolithic reference standards and five sandwich standards are manufactured. To align the components with the NDT requirements for the reference standards, components must only contain artificially inserted defects with a dimensional accuracy of $\pm 1$ mm. Flat bottom holes (FBHs) and double layered polytetrafluoroethylene (PTFE) have been used to simulate delamination defects, whilst inclusion defects are simulated with prepreg backing paper. Existing aerospace standards have been used to define the defect size of $6 \times 6$mm as a minimum detectable defect size [69]. The choice of materials is dependent on processing route, material availability, maintaining simplicity in the absence of additives, and a dual use of references standards to support industry projects. The manufacture of the reference standards was conducted in laboratory conditions of between 18 °C at 63% humidity and 24 °C at 43% humidity, through either quasi-isotropic prepreg hand lay-up or RTM procedures, shown in Figure 5 [69]. The machining was conducted to a $\pm 0.1$ mm tolerance. Details from the manufacture of two monolithic components are summarised with the details of the inputs and processing methods of example reference components in Table 2. The design of component M1 is shown in Figure 6.

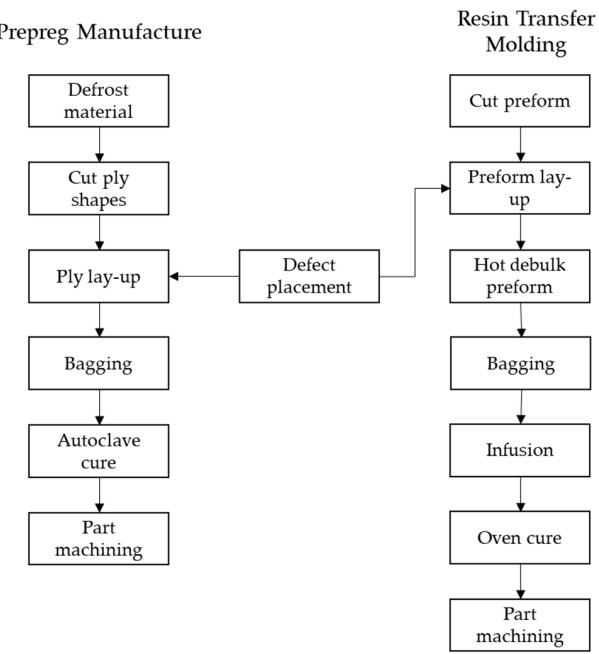

**Figure 5.** Manufacturing process flows for monolithic architecture components.

**Table 2.** Reference component configuration, machining to $\pm 0.1$ mm tolerance.

| Part | Material | Thickness | Processing | Defect |
|------|----------|-----------|------------|--------|
| M1 | CFRP prepreg | Stepped at thickness: 5 mm, 10 mm, 15 mm, 20 mm, 25 mm, 30 mm | Hand lay-up, autoclave (180 °C) | Artificial delaminations, FBHs at near, mid, and far surface |
| M2 | CFRP dry fibre | Stepped at thickness: 5 mm, 10 mm, 15 mm, 20 mm, 25 mm, 30 mm | Resin transfer infusion, oven cure (80 °C) | Artificial inclusions, at near and mid surface |

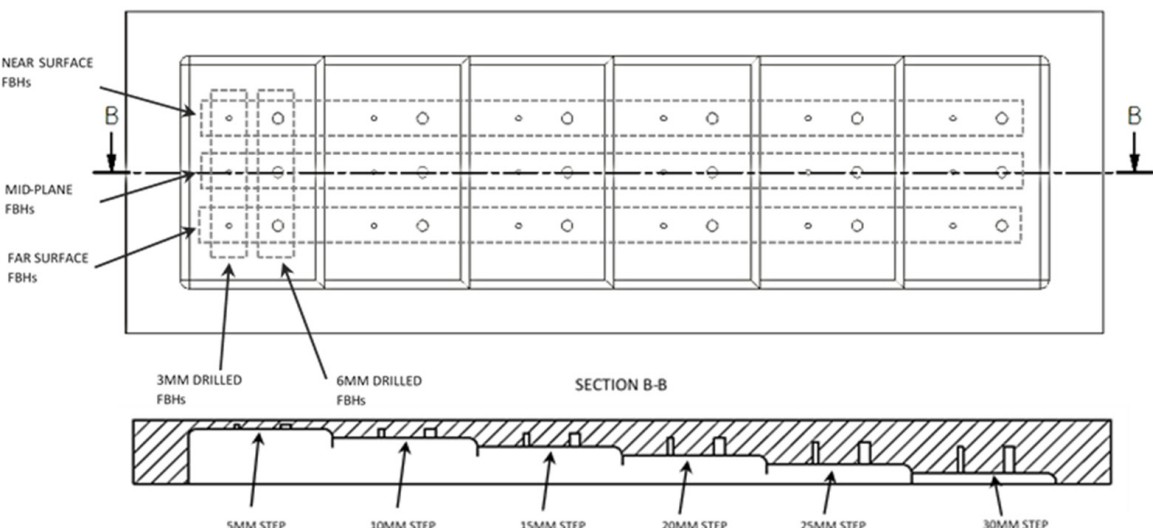

**Figure 6.** Drawing of reference component M1 (first angle projection), plan and cross-section view.

A process capability measurement is undertaken to assess the current capabilities of NDT methods through repeated experimental testing according to standard NCC procedural requirements. UT is one of the most universally applied methods in a variety of industries for the detection of defects; therefore, it is used to baseline the KB population in this framework [70,71]. The procedure for UT methods, and the encompassed techniques, are described: an equipment physical and variable setup is established, and calibration

performed; KPVs are adjusted once the probe is coupled to the tested component and inspection is completed. Within the complex system, all variables are adjusted to produce an 'idealised' response; however, it is recognised that this is highly dependent on the diligence of the NDT operator. Therefore, a degree of human error is assumed [27]. These variables are captured, and the results analysed, noting artificial defects with respect to the as-designed features.

### 3.3. Analyse

Assessment criteria are needed to evaluate the data from an NDT inspection and enable the conversion of knowledge from tacit and complex explicit forms to a qualitative and accessible format. A modified Red–Amber–Green (RAG) rating criteria has been used to assess the data and has been formulated from semi-structured interviews with six NDT specialists, concerning the 'success' of the inspection process in detecting the prescribed defects within the reference specimens. The inspection success is evaluated against two requirements:

- Detection: an anomaly is identified as being present in a structure.
- Characterisation: features of the anomaly are identified and include the size and depth of defects confirmed within a given tolerance.

Tables 3 and 4 demonstrate the evolution of the RAG assessment from an overview of the capabilities suitable for general engineering use, to an expanded set that is more beneficial for technical users. This expanded 'Amber' section was identified as a need for deciding the applicability of the technique based on the suitability of the inspection requirements. For example, it was determined that detection only is more useful in NDT processes than any element of characterisation only. Each inspection result has been analysed against the technical user criteria and entered into the capability matrix along with the reporting of KPVs and defect detection capabilities. An example of population for component M1 and the associated matrix identifiers (IDs) defined by thickness (t) is shown in Table 5.

**Table 3.** RAG rating definition from general user perspective.

| Rating | Definition |
|--------|------------|
| Red | Technique not capable of detection and characterisation |
| Amber | Technique capable of limited detection or characterisation |
| Green | Technique capable of detection and characterisation |

**Table 4.** RAG rating definition from technical user perspective.

| Rating | Definition |
|--------|------------|
| Red | Technique not capable of detection and characterisation |
| Orange | Technique capable of characterisation only |
| Amber | Technique capable of limited detection or characterisation |
| Yellow | Technique capable of detection and part of characterisation |
| Green | Technique capable of detection and characterisation |

**Table 5.** Assignment of RAG rating for UT inspection of component M1.

| Matrix ID | Thickness Band | Matrix Rating |
|-----------|----------------|---------------|
| F1 | $0 \text{ mm} < t \leq 5 \text{ mm}$ | Green |
| F2 | $5 \text{ mm} < t \leq 10 \text{ mm}$ | Green |
| F3 | $10 \text{ mm} < t \leq 15 \text{ mm}$ | Green |
| F4 | $15 \text{ mm} < t \leq 20 \text{ mm}$ | Green |
| F5 | $20 \text{ mm} < t \leq 25 \text{ mm}$ | Green |
| F6 | $25 \text{ mm} < t \leq 30 \text{ mm}$ | Green |

*3.4. Improve A*

Based on the results of the capability matrix and metrics for reliability, a root cause analysis (RCA) is conducted to determine the possible reasons why a detection method is not capable of achieving a 'Green' RAG rating when inspecting a component configuration. The assignment of an expanded 'Amber' or 'Red' rating indicates that an inspection procedure failed to fully detect and characterise a defect, demonstrating that there are inherent risks to the process that may not have been fully explored in an industrial context. In other words, ineffective inspection methods are potentially being deployed, resulting in possible false readings, and impacting the robustness of quality control measures.

To assess the risks in UT techniques, outputs from the RCA are entered into a Failure Mode, Effects, and Criticality Analysis (FMECA). The risks are categorised in four groups: defect configuration, component configuration, human factors, and NDT process system. A selection is shown in Table 6. An FMECA population was conducted through semi-structured interviews with 12 UT composites specialists from a variety of backgrounds and experience in a combination of research and industrial NDT. Each risk was allocated a numerical assessment against two criteria:

- Defect detection (D): Will the failure mode affect the ability to detect and characterise a defect of minimum acceptable size within a part?
- Impact (I): Can the failure mode be recognised and adjustments be made for it during inspection?

**Table 6.** A selection of risks in the FMECA register categorised in four groups.

| Defect Configuration | Component Configuration | Human Factors | Process System |
|---|---|---|---|
| Defect stacking | Material attenuation | Data acquisition | Equipment performance |
| Defect type | Single curvature | Data evaluation | Probe configuration |
| Porosity | Surface roughness | KPV setup | Component movement |

The numerical assignment is shown in Table 7. Calculation of the risk priority number (RPN) is given for the pulse echo UT technique according to:

$$RPN = I \times D, \tag{1}$$

**Table 7.** RPN assessment criteria for FMECA.

| Detection Rating | Detection Definition | Impact Rating | Impact Definition |
|---|---|---|---|
| 1 | Detection and some characterisation possible | 1 | Recognised and can be accounted for in inspection |
| 2 | Detection possible | 2 | Recognised but cannot be accounted for in inspection |
| 3 | Uncertainty in detection, false readings possible | 3 | Not recognised and leads to violation of acceptance criteria |
| 4 | No detection and characterisation possible, false negatives | | |

This value is indicative of the criticality of the inherent risk which may have a certifiable impact on safety as a result of it going undetected. A plot of the RPNs of all risks within the FMECA is shown in Figure 7. Distribution of the results can be evaluated through descriptive statistic measures and visualisation of the results. To highlight the critical failure modes, a method described by Catelani et al. [72], which uses statistical methods to evaluate the 'as low as reasonable practicable' threshold, is used over the Pareto model; the Pareto model exhibits a very conservative assessment of risks. With the adopted method, seven critical failures are identified for high priority resolution.

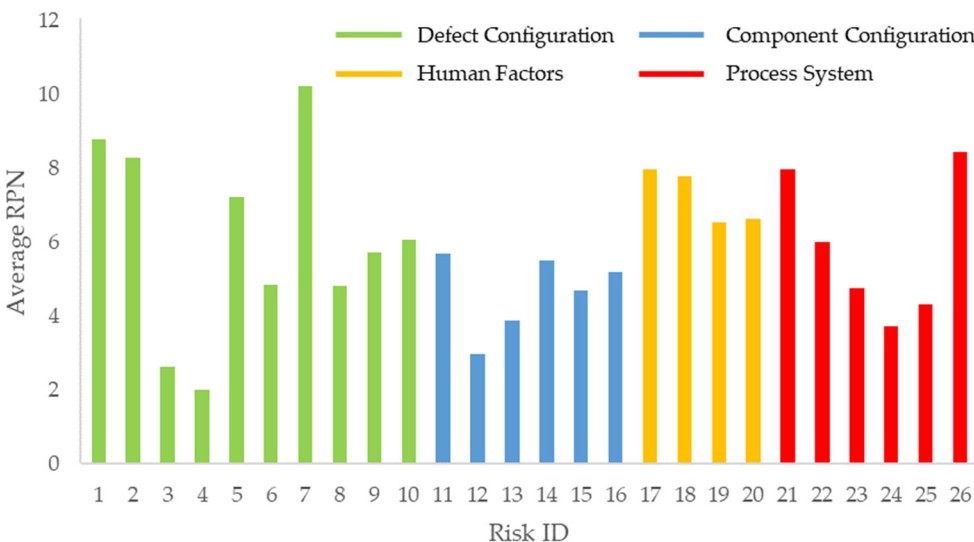

**Figure 7.** Representative FMECA results of risks using RPN assessment criteria.

As the FMECA does not take a combination of failure modes into account, an interaction matrix is proposed to assess the two-dimensional interaction between risks. It is observed that some variables have limited influence on other aspects of the system; however, there are some critical risks that represent a degree of interlinkage that is indicative of a complex system. Combining these results with the FMECA, null hypothesis testing is conducted to assess the RPN, impact of risks, and their interaction with others to determine a route to mitigation.

*3.5. Improve B*

As recorded in the FMECA, whilst a small number of risks were included under human factors, this group had the largest average RPN. Incorporating the industrial requirement to automate the inspection process and the desire to reduce operator-driven risk, a Poka-Yoke analysis is used to evaluate the effect of geometry on automated inspection systems.

Common indicators of complex geometries include sudden changes in curvature, a mismatch in face geometry, and access issues. As such, the surface normal along a scan path can vary drastically from one end of the part to the other; therefore, inspection planning is of great importance to ensure all areas of the part can be reached. For UT inspections, maintaining a surface normal of the probe or water column to the part face is critical; investigations have proven that deviations of up to $\pm 3.5°$ can significantly compromise the response signal. Therefore, variations of the surface normal angle along a scan path are determined as translatable measures of geometrical complexity that will affect the inspection complexity.

Two approaches have been taken to assess the varying complexity: a mesh-based MATLAB method, and a geometrical-based Excel method. Key findings have been evaluated from the geometrical-based model, where a line geometry is selected in CAD, meshed at 1mm intervals, and element normal analysed to compute the angle variation and rate of change of angle along a scan line. With this data, it is possible to select a scan path with the least drastic changes in scan angle and identify probe manoeuvrability problem areas. Figure 8 shows three outputs of the scan paths over a complex hemiellipsoid geometry with a 20 mm radius from the shape to a flat plane. The angle deviation from normal is at 1mm intervals in Figure 9, where $0°$ is representative of normal to a horizontal plate. The rate of change of angle, in Figure 10, is expected to greatly affect the amount of robotic end effector movement needed to maintain a surface normal condition. Using this graph, a preliminary assessment suggests that regions with high peaks and large peak bases will require further analysis to estimate the significant effect of the geometrical deviation.

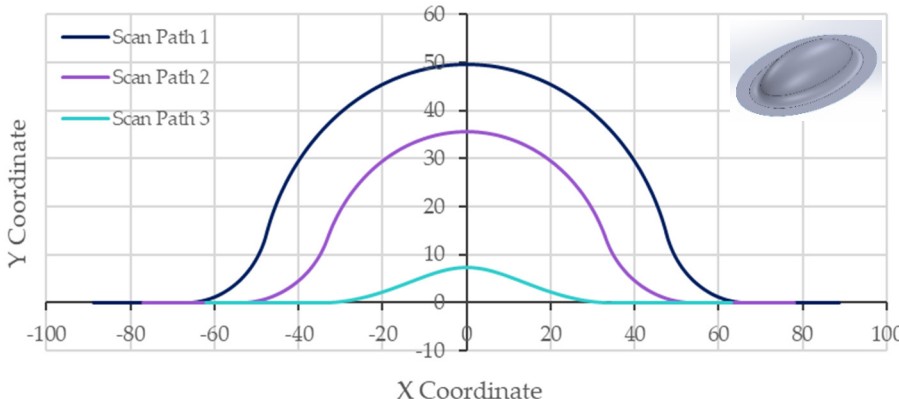

**Figure 8.** Three selected scan paths over a hemiellipsoid shape of 20 mm shape-to-plane radius.

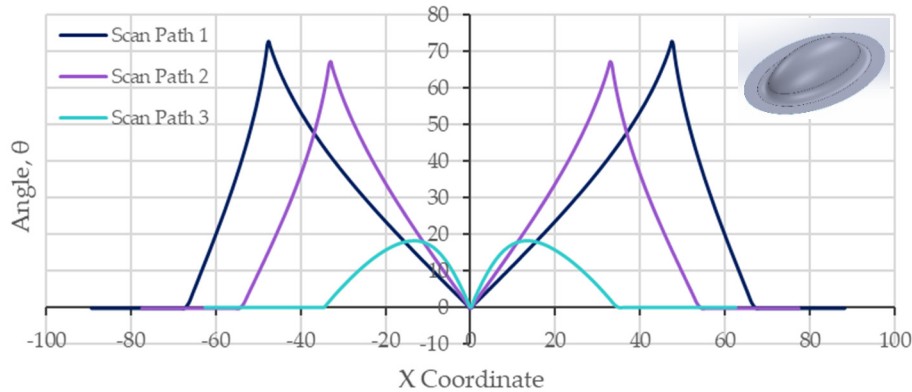

**Figure 9.** Angle variation from surface normal along the scan path x-coordinate.

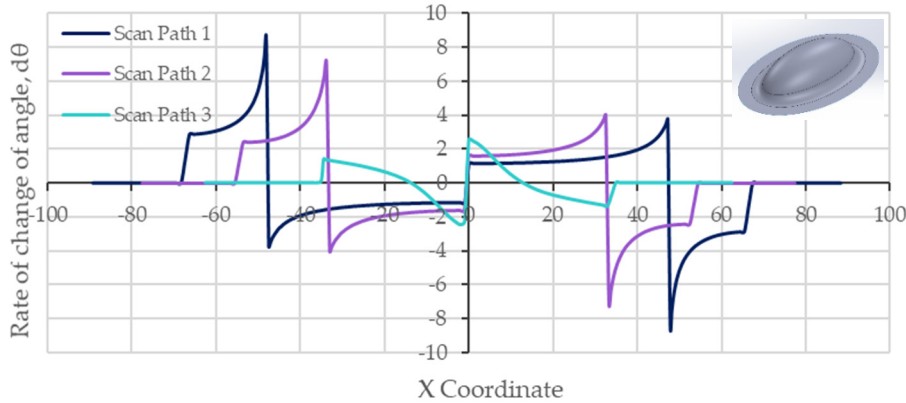

**Figure 10.** Rate of change of angle variation from surface normal along the scan path x-coordinate.

### 3.6. Verify

For the data contained in the KB to be used as a basis for decision making for the deployment of NDT techniques, it must be verified. To improve the robustness of the UT data, gage repeatability and reproducibility (Gage R&R) testing of the scans was completed to understand the variation in the results of an identical setup. Additionally, it provides an opportunity to determine a possible root cause of the missing or mis-characterisation of defects.

Inspection procedures for monolithic components were automated with repeatable scan paths and initiation points for the UT techniques, and data captured at 1 mm increments in an x–y coordinate frame. For each grid point, the data has been averaged against five repeats with the standard deviation computed and normalised, as demonstrated for

component M1 with a pulse echo UT in Figure 11. The initial conclusions have highlighted some trends in variability:

- Variability in ultrasonic signal amplitude increases as the depth increases.
- Scattering is dependent on material quality, and the component and defect geometry. It has been observed that non-uniform materials and curved profiles lead to higher levels of variability.

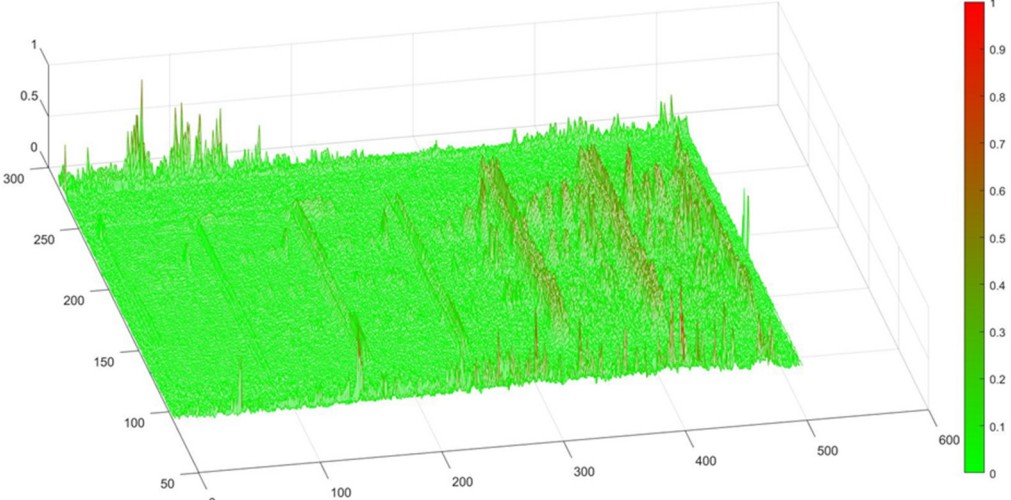

**Figure 11.** Normalised standard deviation of amplitude variation across component M1.

To further gain an understanding of how material composition affects the data produced in UT inspections, monolithic samples were inspected using X-ray computed tomography (XCT) methods. Some correlation in the voidage volume within a specimen is seen, with a higher void content in the dry fibre specimens as compared to the prepreg material specimens.

## 4. Discussion

The adoption of a LSS method enables a systems-based approach to address the NDT knowledge deficit. This framework is used over other improvement methodologies, including total quality management and business process re-engineering due to its flexibility. Additionally, this improvement study exists outside of a production environment and does not directly impact on the improvement of a technical aspect of the inspection process. The tools have been selected and placed within the modified improvement framework to prioritise the key issues and develop effective solutions.

### 4.1. Define

Evaluation of the composite components has demonstrated the breadth in variety of materials, geometries, processing parameters, and defects that may occur within the same process. For a KB to be best utilised, it should contain as many of these configurations as possible such that it can be used as an informative decision tool. With the definition of 34 IDs, it can be argued that the KB is limited in this state; however, the down-selection of component configurations is a necessary step to evaluate the robustness, refine the experimental procedures, and establish the KMS with resource constraints before widespread deployment. Baselining with known quantities will reduce the risk of accepting uncharacteristic inspection method responses, which may appear with complex additions to the component. Once the matrix population process has been proved out, matrix entries can grow to include the system and geometries that are more representative of industrial components.

In manufacture, both the material and component geometry are created during processing, adding complexity in the pursuit of right first time. The defect likelihood is relatively high due to the inherent variability within all stages of the manufacturing process, as discussed by Potter [7]. As such, an assumption should be considered that no two components are completely identical which brings the validity of matrix entries in comparison to 'real' components into question; therefore, validation procedures covered in the Verify phase are important to ensuring the capability matrix remains robust.

Only fundamental and widespread inspection techniques have been selected for the matrix entry; advanced acquisition and processing algorithms have not been utilised. Limited literature currently exists on the KPVs that affect the inspection processes and have been obtained through practical demonstration in this project. It is not certain whether these variables are common knowledge and widely communicated during training such that further documentation is not required. Alternatively, the optimisation of KPVs has been neglected in the research; definition in the capability matrix provides an opportunity to codify this explicit and tacit knowledge.

### 4.2. Measure

Inspection and measurement procedures used in this study are based on the currently documented aerospace standards as typically used by the NCC, a representative of the composites industry, with the component measurements repeated five times to reduce the likelihood of anomalies entering the matrix. Since the reference standards aim to produce 'best-case' scenario products to populate the matrix, defect configuration within the specimen does not allow for a conservative estimate of the defect detection capabilities. Defects in the 'real' components are unlikely to be regular in shape and in unknown locations; the results are highly subject to operator judgement and interrogation persistence. There may also be contention over what constitutes a real defect in comparison to a simulated defect, particularly if the chosen artificial defect manufacture methods are not completely true to the response gained with a 'real' defect. Moreover, any environmental conditions, such as accessibility or human factors, are not considered in this experimental procedure.

As the component composition may vary as a consequence of processing variability and manufacturing performance, coupled with the use of different equipment and interpretations of what 'good' looks like, it is possible that KPVs will not remain consistent across all inspections. Despite the changes in component, equipment, and operator characteristics, it is beneficial to capture the KPVs to establish a benchmark for the standardisation of processes.

### 4.3. Analyse

A consensus is reached within NDT circles that one of the most important factors within an inspection process is an understanding of the component to be tested. This includes knowledge of the architecture, expected defect configuration, and processing route. Alongside this, increasing an operator's knowledge and experience in the detection methods is essential since the results are highly dependent on the diligence and individual characteristics of the operator. It is not known to what degree the results of the matrix would vary if a different operator were to conduct the tests and make important data acquisition and evaluation decisions; however, operator-based performance is the only currently standardised method for carrying out these operations, with the automation-based methods not yet fit for purpose.

Other sources of variability exist within inspection recording, where the quality of the results is dependent on the as-manufactured material quality. This is critical where 'good' regions are used to establish a threshold for defect characterisation. Matrix component configurations are differentiated by defect depth; however, proximity to and the interaction of defects with the back surface can be a larger prohibiting factor for detection and characterisation rather than depth for UT. This issue with the placement of defects

within the component is mirrored, where near-surface defects can create a 'shadowing' effect, potentially creating some misconceptions in sizing.

The assignment of capabilities using the RAG criteria aims to remove some ambiguity from NDT processes such that they can be communicated in a more accessible format. There has been a notable variation in specialist opinion and understanding of the ratings which could be attributed to a difference in experience and environment in which they operate. The rating definitions would benefit from additional input from NDT specialists and users to refine meaning, reinforce understanding, and improve applicability. This is necessary since the rating will not always be deployed in the same environment with identical requirements; therefore, ratings will mean something different in each unique application or organisation. The expansion of the 'Amber' category for a technical user is necessary where there could be a multitude of result variations that signifies some uncertainty or limited confidence in the result. This is an important statement as it indicates there may be poor reliability or a lack of experience of the capabilities in this area and requires more focus. Coupled with industry trends and requirements, this RAG assignment provides an initiation point for additional research and development.

### 4.4. Improve A

The FMECA is inherently a subjective method where any conclusions drawn are highly dependent on the participants. Additionally, failure modes, risk assessment factors, and RPNs are subjective and require an understanding of NDT technologies, organisational processes, and industry standards; however, it also highlights the variance in NDT understanding when approached with an identical question and application. This lack of formalised and unified knowledge across the industry is evident through the spread in some of the RPN values, as shown in Figure 12. This figure presents three examples: various cluster patterns (assumed in Risk 1), mostly universally interpreted (Risk 2), and highly variable opinions (Risk 3). The spread demonstrates the discrepancy in NDT knowledge despite the rigorous training and certification requirements for operators. The greatest contributing factor in the discrepancies is expected to be due to the nature of experience and the organisation or industry in which it was acquired.

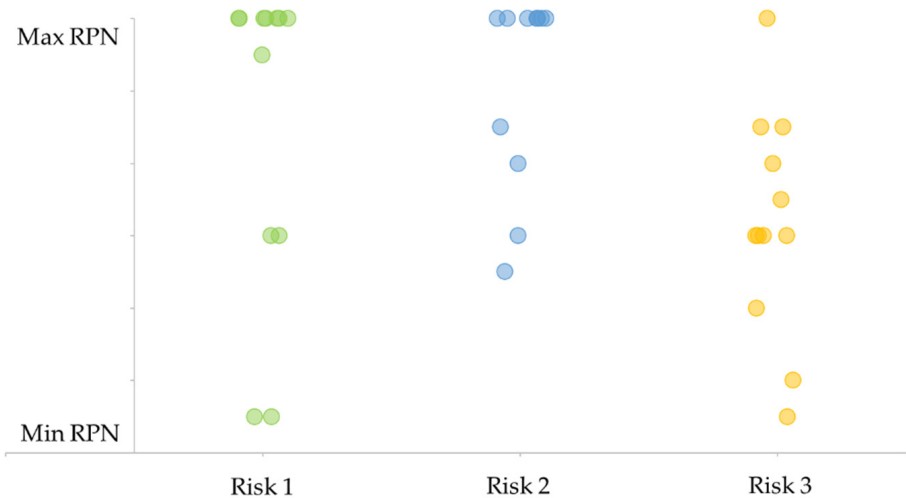

**Figure 12.** FMECA RPN distribution for three risks.

As discussed, NDT is a complex system that involves the interplay of multiple inputs and KPVs. The interaction matrix draws on the relationship between risks, but only accounts for a two-dimensional risk linkage. Multi-dimensional interaction is more likely to occur, where the risks are affected by different degrees; however, simple causal relationships need to be defined first before this matrix can be expanded to encompass multiple risks. As with the FMECA, it is suggested that data collection and entry should be continued to

enable a set of more robust conclusions to be drawn regarding the route to risk priority and resolution.

*4.5. Improve B*

The rate of data acquisition by an automated scan unit remains constant and it is dependent on the distance travelled by the end effector as opposed to acquisition as a function of time; however, the velocity of the end effector will vary according to the movement required by robotic manipulators to access the component geometry. It is assumed that for complex geometry or regions where the geometry changes significantly over a small area, the manipulators must move much more than over a flat surface. Consequently, the end effector head will reduce in velocity. Understanding how the geometrical profile of components will affect the UT end effector movement is critical in adding intelligence to automated inspections through defining the acceptable surface normal deviations. Moreover, the rate of change is an essential characteristic for evaluating the productivity of a scan path.

Some trends in the data have shown that for any feature that deviates from $0°$ from a surface normal creates some complexities in path planning; however, a tighter radius over the same travelled angle is more likely to create a disturbance in the velocity of the scan path. Similarly, with the same radius and a different travelled angle, a higher angle will result in a longer end effector journey time due to a higher deviation from $0°$. This model treats the geometrical features as discrete objects and does not account for the impact of feature succession. Additionally, the model does not allow for an assessment of the inspection efficacy as it does not take the quality of the inspection into account. As such, the correlation between the direction of scan path and quality of data output for multiple directional paths will need further development and analysis.

*4.6. Verify*

Gage R&R testing is necessary to evaluate both the inspection results in the capability matrix, and the factors affecting the measurement system variation; however, it has been observed that it is not just measurement system intrinsic capability, application parameters, and human factors that will affect the inspection result. Manufacturing variability may also affect the reliability of data acquired through inspection methods; it should be considered what the structural integrity of a reference standard looks like. The intended use for the standards will determine whether it should present as a best-case scenario for establishing the equipment capability or have characteristics with potential non-designed features that are representative of the component to be tested.

Inspecting with XCT methods is considered the 'optimum' method for quality interrogation; however, due to equipment, computational, and size limitations, it presents some limitations in the inspection of all components. Whilst the integrity of the reference specimens used in the capability matrix was validated through this method, it raises some questions as to whether this method is necessary for all inspections. Along with the varying requirements for inspection methods, XCT has reinforced the results of other UT testing; it must be evaluated in each use case if this expensive method should be applied where others will provide similar results.

*4.7. Impact Assessment*

When coupled with changing industry requirements, the KB has the potential to assist in road-mapping the development of techniques through resource prioritisation to reach a capability threshold. Whilst other KB are in existence within NDT circles, they are either restricted to a select group or have been rendered obsolete. In a study that ran from 2016 to 2019, the BINDT conducted a round robin exercise with various academic institutions and industrial companies in which manufactured reference specimens with artificial fibre discontinuities were tested. Out of the nine organisations contacted, three were able to provide the results obtained from testing. As expected, thoroughness in the inspections and

results varied considerably. Whilst this exercise demonstrated an attempt to capture the state-of-the-art, the commitment to following through with collecting and communicating the data has waned. As such, it has not contributed to improving the collective NDT knowledge. The KB developed in this study provides a more robust improvement framework that has been established and proven out with selected component configurations and detection methods. Whilst baselined with UT, further development and iterations of the KB population can be conducted to expand the scope of content, including more widespread component configurations and inspection methods. Additionally, exploration and consultation with NDT specialists has been conducted to understand the needs of NDT operators and incorporate their voice into the knowledge communication.

Exploiting the KB on inspection operations introduces an element of intelligence; the organised information contained with the KB supports evidence-based inspection plans for the improvement of NDT operations. The capability matrix can be used to underpin the effectiveness of an NDT system, providing the rationale to support the application of a certain method during inspection through a delegation tool. This is critical when utilising an automated system, such as CIVC, where knowledge-based applicability data should be used to make key decisions in operations. Whilst automation removes an element of variability, it introduces further complexity to the process and demands a comprehensive understanding of the fundamental capabilities of the interaction between automation and detection methods. The preparation and parameter selection for inspection of a new component using the KB could significantly reduce this pre-inspection setup whilst robustness, efficacy, and productivity can be assessed in inspection plans. Continued learning with knowledge capture will contribute to eliminating the weak links or unknowns in operations.

The KB provides a key enabler for the introduction of KM activities into NDT. In developing the KB into a KMS, the capability matrix will need to integrate with other phases of the product introduction process. This initiates the steps to create a DFI methodology; a feedback loop can be enabled between the composites design, manufacturing, and inspection to augment decisions and manage expectations. Validation data from the NDT provides assurance for the optimisation of components, with the knowledge that quality controls will be effective. Moreover, the design of components can be achieved with the understanding of what the inspection methods are capable of and which ones can be employed.

## 5. Conclusions

This study has demonstrated the steps taken in the development of a KB using a modified LSS Define–Measure–Analyse–Improve–Verify framework to support research and industrial goals for NDT. Demonstration of the inspection system complexity contributes towards the attainment of two aims:

- The development of KM activities for the documentation of inspection results and technical know-how in the form of a high-fidelity KB.
- Establishes a framework for exploiting NDT knowledge to inform the design for the inspection of components.

A framework towards the generation of a KB capability matrix, mapping the current state capabilities of selected NDT techniques to component configuration demonstrates the creation of a knowledge resource. A capability matrix is constructed using the key inputs to the inspection process, evaluated from a SIPOC diagram. KPVs from the inspection process of reference specimens have been captured, and the results analysed for the efficacy of detection and characterisation of as-designed artificial defects. This efficacy is evaluated against RAG assessment criteria and assigned a rating in the population of the matrix. Failures to detect and characterise defects are captured through RCA in an FMECA with a view to identify the potential routes to a resolution. Some operator-driven failures are removed from the system in the study of automated path planning, correlating geometrical complexity to the possible difficulties in NDT. For robustness of the KB, a Gage R&R

assessment was completed on UT matrix data. Through the integration of the KB in a DFI methodology, deployment in industry will drive the improved intelligence of inspection operations, capability and productivity gains, and support the transition towards NDT 4.0 in the modern composite manufacturing environment.

**Author Contributions:** Conceptualisation, N.G.; methodology, N.G.; validation, N.G.; formal analysis, N.G.; investigation, N.G.; resources, N.G.; data curation, N.G.; writing—original draft preparation, N.G.; writing—review and editing, N.G. and C.W.; visualisation, N.G.; supervision, R.R., A.J.C. and C.W.; project administration, N.G., R.R., A.J.C. and C.W.; funding acquisition, N.G., R.R., A.J.C. and C.W. All authors have read and agreed to the published version of the manuscript.

**Funding:** This research was supported by the Engineering and Physical Sciences Research Council through the EPSRC Centre for Doctoral Training in Composites Manufacture (grant: EP/L015102/1); and The Future Composites Manufacturing Hub (grant: EP/P006701/1).

**Data Availability Statement:** All data necessary to reproduce the results and support the conclusions are included in this paper.

**Acknowledgments:** The technical and financial support provided by the National Composites Centre is appreciated. The authors acknowledge the μ-VIS Imaging Centre at the University of Southampton for provision of tomographic imaging facilities, supported by EPSRC grant EP-H01506X.

**Conflicts of Interest:** The authors declare no conflict of interest.

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
