# Peer review of "Understanding System Complexity in the Non-Destructive Testing of Advanced Composite Productsâ€"

_jmmp, doi:10.3390/jmmp6040071_

Round 1
Reviewer 1 Report
The manuscript is centered on the complexity in using NDT on composites. It is an important issue, regarding the increasing use of composites and the need to improve parts reliability and thrust.
Some amendments can be implemented, as follows.
In section 2 Current State, authors describe Ultrasonic Testing and Thermographic Testing. The latter was never used along the text and there is an add-on, X-Ray Computed Tomography (page 14) that deserves some description. Did authors consider other NDT methods, like TeraHertz or Eddy Current? It would be interesting to have that in the manuscript.
Also, numbering of subsections need to be corrected, as the following example - 3.1 Ultrasonic testing should be 2.1, the same applies to the remaining section 2.
The numerous methodologies referred to at the text, like DMAIV, SIPOC, RAG, FMECA, RPN, etc, were defined by authors or were based on precedent work from the same/other authors? Please clarify.
What was the rational for the design of M1 and M2 parts?
In section 4.2, authors refer to the use of aerospace standards, which standards as they are not identified?
In figures 8, 9 and 10, the color code is the same as in figure 7? It is not clear to the reader.
Please check spelling, as there is some "to addressing" (line 480 as example of "to verb+ing") or similar along the text, I believe it should be "to address".
Author Response
Dear reviewer,
Thank you for your comments on this paper. In response to the points you raised, I have made the following amendments, all of which can be found by additions in the manuscript.
1. In section 2 Current State, authors describe Ultrasonic Testing and Thermographic Testing. The latter was never used along the text and there is an add-on, X-Ray Computed Tomography (page 14) that deserves some description. Did authors consider other NDT methods, like TeraHertz or Eddy Current? It would be interesting to have that in the manuscript.
This selection of NDT methods has been conducted to prove out the matrix are based on an established and an emerging method. This is a first base to establishing the baseline of the framework before other methods can be added to the matrix. In the paper, exemplar population is performed with Ultrasonic Testing only.
2. Also, numbering of subsections need to be corrected, as the following example - 3.1 Ultrasonic testing should be 2.1, the same applies to the remaining section 2.
Numbering corrected.
3. The numerous methodologies referred to at the text, like DMAIV, SIPOC, RAG, FMECA, RPN, etc, were defined by authors or were based on precedent work from the same/other authors? Please clarify.
Whilst the DMAIV framework has been modified by the authors, the other methodologies are common and well established across LSS activities. A comment has been made to acknowledge this.
4. What was the rational for the design of M1 and M2 parts?
Comment has been made on how the design of reference components has been informed by making them representative of components within the composites industry.
5. In section 4.2, authors refer to the use of aerospace standards, which standards as they are not identified?
Unable to identify the particular standard from which the process originates however it is used by the NCC, a research institute that is representative of the 'ideal' standard for composites manufacturing. A comment has been made in the text.
6. In figures 8, 9 and 10, the color code is the same as in figure 7? It is not clear to the reader.
Figures 8, 9, and 10 have been modified to avoid confusion.
7. Please check spelling, as there is some "to addressing" (line 480 as example of "to verb+ing") or similar along the text, I believe it should be "to address".
Manuscript has been proofread to try eliminate errors.
I hope these amendments are to your satisfaction.
Kind regards.
Reviewer 2 Report
This paper deals to an issue very important in the NDT sector in general, relating to the evaluation of their quality results.
For this aim, it is considered the development of a Knowledge Based system including the complexity in NDT measurements for composites. The construction of a matrix with all the inputs affecting the inspection process is carried out for an artificial defects design. The efficacy is evaluated although some aspects are not considered as geometrical or operator failures.
However, it is not clear if the study is valid for any NDT or it is just checked for Ultrasonic. Furthermore, thermography and X-ray are mentioned but the last just as verify method, although it is also a NDT methodology. This aspect could be clarify in the text.
Anyway, it is derived that further studies are necessary to adjust the methodology proposed but it constitutes a robust one.
Author Response
Dear reviewer,
Thank you for your comments on this paper. In response to the points you raised, I have made the following amendments which can be found by additions in the manuscript.
...it is not clear if the study is valid for any NDT or it is just checked for Ultrasonic. Furthermore, thermography and X-ray are mentioned but the last just as verify method, although it is also a NDT methodology. This aspect could be clarify in the text...
The selection of NDT methods has been conducted to prove out the matrix are based on an established and an emerging method. This is a first pass to establishing the baseline of the framework before other methods can be added to the matrix. In the paper, exemplar population is performed with Ultrasonic Testing only. Comments within the research activities and discussion sections address this concern.
I hope these amendments are to your satisfaction.
Kind regards.
This manuscript is a resubmission of an earlier submission. The following is a list of the peer review reports and author responses from that submission.